# Tail and Spinal Cord Regeneration in Urodelean Amphibians

**DOI:** 10.3390/life14050594

**Published:** 2024-05-07

**Authors:** Eleonora N. Grigoryan, Yuliya V. Markitantova

**Affiliations:** Koltzov Institute of Developmental Biology, Russian Academy of Sciences, 119334 Moscow, Russia; leonore@mail.ru

**Keywords:** amphibians, tail regeneration, spinal cord regeneration, morphogenesis and patterning de novo, molecular players, external factors

## Abstract

Urodelean amphibians can regenerate the tail and the spinal cord (SC) and maintain this ability throughout their life. This clearly distinguishes these animals from mammals. The phenomenon of tail and SC regeneration is based on the capability of cells involved in regeneration to dedifferentiate, enter the cell cycle, and change their (or return to the pre-existing) phenotype during de novo organ formation. The second critical aspect of the successful tail and SC regeneration is the mutual molecular regulation by tissues, of which the SC and the apical wound epidermis are the leaders. Molecular regulatory systems include signaling pathways components, inflammatory factors, ECM molecules, ROS, hormones, neurotransmitters, HSPs, transcriptional and epigenetic factors, etc. The control, carried out by regulatory networks on the feedback principle, recruits the mechanisms used in embryogenesis and accompanies all stages of organ regeneration, from the moment of damage to the completion of morphogenesis and patterning of all its structures. The late regeneration stages and the effects of external factors on them have been poorly studied. A new model for addressing this issue is herein proposed. The data summarized in the review contribute to understanding a wide range of fundamentally important issues in the regenerative biology of tissues and organs in vertebrates including humans.

## 1. Introduction

The extraordinary regeneration abilities of caudate amphibians (Urodela, family Salamandridae) have attracted researchers’ interest for many decades. Dedicated studies have considered various aspects of this process: morphological, cellular, evolutionary, genetic, and molecular [1,2,3,4]. Besides the well-known ability to regenerate limbs, these animals, unlike mammals, are capable of very close to full regeneration of the spinal cord (SC) with the recovery of functional neural links [5,6]. Salamanders can regenerate the tail and the SC after amputation or damage repeatedly and throughout their lives [7,8,9]. There are great differences in cellular and molecular regenerative responses between regeneration-competent amphibians and the SC of mammals, including humans, which is not capable of regeneration. Salamanders are distinguished by the rapid activation of source cells of regeneration of the SC and the entire tail, by the specific method for healing the damage and restoring the organ, and also by the relationship of the regenerating SC and other tail tissues. All the regenerative responses in salamander’s tail and SC are aimed at initiating and progressing the epimorphic regeneration process [10]. In mammals, it is vice versa; the extensive death of neurons and glial cells occurs after SC damage, followed by local healing of the SC through the formation of glial scar at the damage site [11]. The scar formation inhibits the entire set of regeneration processes, SC growth and functional recovery, as well as axon growth [12,13]. The blockage of axon regeneration caused by scar tissue at the site of injury causes the loss of neural links and deficiency of function. Attempts to restore the regenerative potential in mammals have been made for other mammalian tissues [14,15,16,17,18] and for the SC [13,19,20,21,22,23]; however, these attempts have not brought much success. This explains the relevance of the present study of the cellular and molecular factors contributing to the regeneration of the SC and tail in salamanders in general. The studies above provide a path for understanding and developing approaches to regenerative SC therapy that would facilitate structural and functional SC regeneration in mammals including humans.

In Urodela, the SC regenerates in cohort with other tail tissues. Tail amputation in urodelean amphibians, performed to induce regeneration, is a more frequent choice of operation compared with local damage (crush) to the SC [24,25,26,27,28]. It is obvious that the model of tail amputation together with an SC fragment within it is not consistent with cases of natural (in humans) or artificial (in animal models) spinal column and SC injury. For this reason, the direct extrapolation of the information discussed here to amniotes currently requires great caution and further accumulation of modern information obtained in comparative studies.

Histogenesis and morphogenesis in development and regeneration largely obey the self-organization laws, including intrinsic genetic development programs [29,30,31]. In parallel, there are cumulative influences from the environment and other developing tissues of the body. The molecular participants in these influences are factors secreted by mutually developing surrounding cell populations, systemic factors (hormones and inflammatory factors), and also dynamic physical factors. The mutual influence of tissue mechanics and biochemical signaling accompanies and determines the morphogenesis of organ tissues and its patterning, both in development and during regeneration [32,33].

In a mature organism in which the ontogenetic programs have completed their work, any correct morphogenesis of whole organs de novo is impossible in higher vertebrates. The known exceptions are the extensive healing of the fingertip in children and young mice [34,35,36], the auricle in mice of some lines [37,38], and horn regeneration in deer [39,40]. It is also reported that in spiny mice (*Acomys*), SC healing can be detected after injury in the absence of glial scar formation [41].

The tail regeneration in Urodela is an example demonstrating the possibility of correct morphogenesis in accordance with genetically determined form and functions; an example is the reproduction of morphogenetic processes in development by sexually mature organisms. This makes the model an excellent tool for understanding them.

The regulation of developmental processes by factors that are external relative to the body is also one of the essential issues in developmental biology. Environmental regulation of development can cause different phenotypes with the same genotype to emerge, which contributes to adaptation to the environment [42]. One of the forms of such adaptation is a non-pathological variation in the shape of the outer organ that occurs with its growth and morphogenesis. The regeneration of the tail and SC in Urodela appears to be a convenient experimental model of adaptive morphogenetic plasticity during regeneration. Nevertheless, it should be noted here that the phenomenon of effects of environmental factors on the regenerating tail morphogenesis has not been investigated. A rare example of external effects at the level of morphological alterations and molecular regulatory signals is represented by the changes in the shape of the regenerating tail and SC in Urodela exposed to conditions of gravity dose variations [43,44] and heat shock [45].

In the present review, we summarize data obtained using a model of SC and tail regeneration in urodelean amphibians (axolotls and newts). These data concern the general course of the regeneration process and the regulatory mechanisms at its various stages. A special section is provided to consider morphogenetic changes in the regenerating tail under the effect of external factors.

It should also be mentioned here that a vast number of recent studies have been carried out on other models for appendage regeneration: regeneration of the fin and SC in zebrafish [46] and of the tail in cephalochordates, commonly called amphioxus or lancelets [47], as well as caudate amphibians (Anura) [48] and reptiles such as lizards [49,50,51] and geckos [52]. These studies are widely presented in recent reviews [46,51,53,54,55]. We take these data into account, but the focus of the article is mainly on tail regeneration in salamanders.

## 2. Pedomorphosis: Pattern of Development in Caudata Amphibians

Urodela amphibians (family Salamandridae) comprise the so-called caudate amphibians. A characteristic feature of these animals is a pedomorphic state, which, in turn, results from heterochrony [56,57]. The heterochrony leads either to a delay in somatic development with the normal maturation rate (neoteny) or an accelerated process of sexual maturation, after which the somatic development is not fully completed (progenesis). In salamanders, pedomorphosis has been identified in both forms and has been shown to be regulated by a thyroid hormone (TH) in both cases [57,58,59]. The existing explanations for the high regenerative capacity in salamanders are associated with pedomorphosis as a key ontogenetic factor permissive to regeneration [9,60]. It is also known that pedomorphosis correlates with an increase in genome size, which, in turn, positively correlates with cell size but negatively correlates with the rate of cell proliferation and differentiation [61].

## 3. Major Course of the Tail and SC Regeneration Process

### 3.1. Wound Healing and Blastema Formation

After amputation of appendages, a healing process initially occurs that, in general, resembles the healing of epidermal tissue [62,63,64]. During the first hour post operation, bleeding stops, a fibrin clot forms, and the integumentary epidermis migrates onto the amputation surface. Then, the processes associated with the removal of damaged cells begin: Edema, phagocytes, and other cells involved in the inflammatory reaction penetrate into the stump, osteocyte death occurs, etc. [65].

When the wound surface is closed, the dedifferentiation and proliferation of source cells of regeneration, produced by stump tissues and poorly differentiated progenitor cells, are initiated under it [7]. In the classical definition, blastema of appendages is a mass of undifferentiated, mesenchymal cells formed through the interaction of stump cells with wound epidermis and also due to signals from the peripheral nerves [7,66,67,68].

Morphologically, blastema cells are a pool of uniform blast cells, proliferating progenitor cells of regenerate’s tissues [69] (Figure 1).

There are various hypotheses about the origin of blastema cells of appendages. According to them, blastema cells can be originated through dedifferentiation of damaged tissue cells, transdifferentiation of lineage-restricted progenitor cells, and also through proliferation of resident stem cells [70]. One of the early studies, where the triploid/diploid cell marker in the axolotl *Ambystoma mexicanum* were used, considered the issue of the relative extent to which cells of the dermis and skeleton of the stump contribute to the blastema. According to the authors [71], cells of dermal origin make up 43% of the blastemal cell population, and only 2% are related to skeletal tissue. It is assumed that the significant contribution of dermal cells to blastema formation affects the pattern of appendage development de novo, whereas the skeletal tissue has virtually no effect [71]. Currently, efforts are being made to clarify the origin of blastema cells by advanced methods. There have been attempts to estimate the contribution of connective tissue (CT) cells of an axolotl’s stump using single-cell sequencing and lineage tracing [72]. The results of genetic fate mapping combined with scRNA-seq of CT cells have shown that dermal and interstitial CT fibroblasts are the major contributors to the blastema [73]. These fibroblasts dedifferentiate toward common, multipotent CT progenitors that rebuild the regenerate’s patterned skeleton. Such dedifferentiated blastema cells regain an embryonic gene expression program [73]. The issue of source cells for the formation of various tail regenerating tissues is discussed in more detail below (Section 3.3).

One of the differences between the epimorphic tail and limb regeneration in caudate amphibians and the wound healing in higher vertebrates is the formation of wound epidermis. After amputation and wound closure, the epidermis grows thicker and gives rise to the apical epithelial cap (AEC) [65]. Immediately below the epidermis, swelling of tissues occurs, with a large number of red blood cells, dead cells, and debris that appear more clearly on the ventral side. According to [74], the processes of AEC cells provide them with phagocytic activity, which, along with the macrophages’ activity, is important for removing numerous damaged cells. The AEC is regarded by researchers as a kind of signal center that has not been fully studied [75,76]. The term “signal center” implies the presence of active signaling from the AEC, which determines the behavior of the stump cells and the emerging blastema [77].

Blastema cells require the presence of growth and trophic factors for their development. Among the known factors secreted by the AEC and peripheral nerves are members of the FGF family, glial cell line-derived neurotrophic factor (GDNF), substance P, transferrin, etc. They maintain cell viability and regulate their proliferative activity [68]. As shown in the case of limb regeneration in axolotl, AEC cells, in turn, are under regulatory control from the nerves innervating the stump [78]. Thus, interstitial and intercellular molecular interactions occur as early as at the first stage of the regeneration of a salamander’s appendages [7,79] and work in accordance with the positive feedback loop principle [80] (Figure 2).

Of particular note is the challenge of studying the AEC. It is explained by the constant spatiotemporal modulation of the molecular signature of AEC cells and by varying signaling. Furthermore, as the blastema develops, the level of differentiation of its cells varies, and as the SC develops, their signaling also changes.

As blastema cells leave the dedifferentiated state and the reproduction phase, histogenesis is initiated in the tail regenerate. Like histogenesis in any organ, the histogenesis in the tail regenerate is a well-coordinated process of the formation of SC, muscles, cartilage, connective tissue, and integument tissue [24,81,82,83,84,85].

### 3.2. SC Regeneration: General Course

In the tail regeneration process, salamanders can regenerate the functional SC [86]. The process consists of the restoration of the damaged SC region and extension of the SC ependymal tube from the damage site rostrocaudally, with subsequent differentiation of cells into neuronal and supportive ones [87]. The ependymal tube ends with the anterior margin, the so-called terminal vesicle whose cells exhibit a proliferative and migration activity that leads to tubular outgrowth [88] (Figure 2). The ability of cells of the SC regenerate to divide and form the lost fragment of the CS is a necessary prerequisite for building the entire cytoarchitecture of the tail regenerate.

A question has arisen from time to time whether some other reserve cells, in addition to ependymoglial ones, are involved in the SC regeneration in Urodela. To answer it, re-amputations of the tail were made in mature *Triturus carnifex* [89]. After seven amputations and a comparison with the animals that were subjected to a single amputation, it was found that the tail regeneration in repeatedly operated animals did not show changes. The authors suggest that in the case of involvement of SC reserve cells in the regeneration, the latter would be hampered due to the depletion of the cell source, which did not happen in reality [89]. Of particular interest is the fact that capillaries around the regenerating SC are discrete not only in the early regeneration stages but also at two months post amputation. Thus, during the tail regeneration, the blood–brain barrier is not fully formed and is not completely effective, and the possibilities for the exchange of metabolite and growth factor between nervous tissue and blood are great [90].

Thus, shortly after being damaged, the SC forms a population of ependymoglial cells that play a role of neural stem cells (NSCs) capable of differentiation into new glial and neural cells of the SC [91,92,93,94]. Morphologically, these cells are oval, having long neural processes. Ependymoglial cells respond to damage signals by rapid activation and initiation of proliferation [94,95,96,97]. Neural differentiation in cell cultures from regenerating SC occurs not only due to the neurons that were originally present in the explant but also through the active division and differentiation of ependymal cells. From this, an assumption has been made about a bipotent precursor of neurons and glial cells in the ependymal lining of the SC in caudate amphibians [98]. The proliferative activity of ependymoglial cells is dependent on the planar cell polarity (PCP) signaling pathway [94]. PCP represents protein-mediated signaling that coordinates cell orientation [99]. The co-expression of GFAP and the transcription factor (TF) Sox2, a marker of NSCs, is specific to the molecular genetic profile of ependymoglial cells [95,100,101]. The knockdown of the *Sox2* gene, performed on an axolotl’s tail amputation model using the CRISPR/Cas9 technology, was shown to slow down the growth/regeneration of both the SC and the tail regenerate in general [92,101]. Labeling of SC axolotl ependymoglial cells in vivo using GFP, being under the promoter of the GFAP-encoding gene, which allows monitoring of GFP^+^ cells during tail regeneration, showed that most of these cells are indeed producers of neurons and glia of the SC regenerate [91]. However, as was found in this case, a small number of GFP^+^ cells leave the SC regenerate and become involved in the formation of muscles and cartilage of the tail regenerate in 20% and 8% of cases, respectively [91]. This example of switching between cell types shows that at least some SC cells of caudate amphibians are capable of deep dedifferentiation. The results of grafting GFP^+^ labeled fragments of the SC to non-transgenic recipients confirmed this observation [100]. Later on, it was found that in the case of tail amputation in salamanders, FGF and Wnt signaling pathways are the regulators of the behavior and fate of NSCs, while inhibition of one of them blocks regeneration [94,102,103,104]. An assumption was made that ependymal cells of the re-growing SC produce neurotrophic factors that stimulate the survival of neurons and axon growth. These factors include NGF, the brain-derived neurotrophic factor (BDNF), neurotrophin-3 (NT-3), neurotrophin-4/5, (NT-4), the ciliary neurotrophic factor (CNTF), (CDF/LIF), and the cholinergic neuronal differentiation factor/leukemia inhibitory factor (CDF/LIF) [66] (Figure 2).

Once again, we emphasize that salamanders, unlike mammals, do not form a glial scar, which is a barrier to the recovery of axon growth [105,106,107]. In humans, the activation of glial cells that leads to scar formation is accompanied by increased glial differentiation of SC cell phenotypes: reactive astrocytes, NG2^+^ glia, and microglia surrounding the lesion site [11]. These cells express a range of proteins that inhibit/block axonal regrowth such as vimentin (Vim), GFAP, and ECM proteins, including chondroitin sulfate proteoglycans (CSPGs) [11,106,108,109,110,111]. Furthermore, following injury, nerve and glial cells die as a result of activation of microglial cells and infiltration of macrophages and lymphocyte cells [112]. It is obvious that the inhibition of GFAP expression in progenitor cells of the SC and, on the other hand, the stimulation of their neuronal differentiation and viability are the key issues of research.

The study of the recovery of the peripheral nervous system and innervated target muscles during the tail regeneration has also been of interest [67]. The process of axon outgrowth was described in detail using the model of SC transection in the newt *Notophthalmus viridescens* [85]. The growth of transected axons through the damaged area was observed using labeling and imaging techniques. Immediately after damage, the SC axons reduce on both sides of the wound and undergo dystrophy but resume growth after a week. The growth of axons in the damage area is regulated by signals from astrocytes, meningeal cells, and glial cells lining the central SC channel. These cells regenerate first at the damage site and are associated with the loose ECM that allows axon growth cone migration. In conclusion, meningeal cells, axons, and glia move as a unit to close the gap in the SC. Axons sprout from the injury site through white matter, following their functional targets [85].

### 3.3. Regeneration of Tissues Surrounding the SC of the Regenerating Tail

As the tail regenerates, the blastema cells, while actively proliferating, are concentrated in accordance with the initially set pattern, which results in the formation of mesodermal tissues: cartilage, muscles, and connective tissue (CT) [86,113,114]. As mentioned above, studies of blastema of regenerating appendages indicate the heterogeneity of its population. Mesenchymal cells of the stump (dermis, fibroblasts, and spinal cord membranes) and Schwann cells are assumed to make the major contribution to the blastema formation [7]. An assumption was also made that blastema cells are formed in various ways: as a result of dedifferentiation of stump cells and activation of resident progenitor cells [115,116,117,118]. Earlier, to estimate the contribution of stump muscle cells to tail regeneration in urodelean amphibians, Dinsmore [119,120] used unilateral ablation of tail soft tissues with subsequent amputation at a more distal level. It was found that the loss of soft tissues in the tail stump does not have any marked effect on the course of regeneration. The full regenerate developed on an asymmetric stump that lacked muscles on one of its sides. The data on the lack of a direct contribution of the stump muscles to the blastema formation and regeneration raised the question of the source of muscles formed in regenerates [84,119,120]. Nowadays, authors of studies generally agree that muscle regeneration of appendages in salamanders can occur via two main mechanisms: the proliferation of dedifferentiated muscle cells or the proliferation of Pax7+ satellite cells [91,121,122]. It is important to emphasize here that the issues of the origin of appendage blastema cells in salamanders have been studied, to a greater extent, on a limb regeneration model rather than on a tail regeneration one. Nevertheless, the appendages’ (e.g., tails and limbs) regeneration paradigms are considered the same or similar [123]. According to [116], in the case of axolotl limb regeneration, damaged muscle fibers differentiate to form mononuclear cells that make up a noticeable portion (17%) of the population of all blastema cells. It is also known that the dedifferentiation of multinucleated muscle cells of the stump into mononuclear progenitor cells occurs under the control of an evolutionarily conserved “muscle differentiation” gene, *msx1* [124]. Attempts to inhibit the *msx1* expression were made by the morpholino-mediated knockdown method using the model for the regeneration of axolotl’s tail muscles in vivo [125]. However, no marked effect of *msx1* on the process of muscle fiber dedifferentiation has been recorded. It is assumed that the fate of damaged muscle fibers may vary within the stump. As we mentioned above, there is a report on the dual fate of certain ependymoglial cells of the SC during an axolotl’s tail regeneration in vivo [91]. According to these data, the regenerating SC cells are multipotent neuromesenchymal precursors. Some of these cells migrate from the SC region and are involved in the muscle and cartilage formation. Simultaneously, an assumption exists [55,126] that the production of ectodermal and mesodermal precursors by ependymoglial cells of the SC in the axolotl depends on the pattern of the injury and the course of SC regeneration. It is also assumed that the mechanisms of formation of progenitor cells for muscle formation in the case of limb regeneration may vary depending on the species of salamanders. Differences were documented for the newt (*Notophthalmus viridescens*) and the axolotl (*Ambystoma mexicanum*) [121]. The results of Cre-loxP genetic fate mapping showed that the dedifferentiation of muscle fibers is an event characteristic of limb regeneration in the newt but not in the axolotl. In the newt, fragmentation of the original muscle fibers and production of proliferating PAX7**^−^** mononuclear cells occur, which leads to the de novo muscle formation. In the axolotl, myofibrils do not generate proliferating cells, and resident PAX7**^+^** cells are used for muscle regeneration. These results demonstrate significant diversity of limb muscle regeneration mechanisms among salamanders [121]. According to [117], in the newt *Notophthalmus viridescens*, a population of multipotent Pax7**^+^** satellite cells, localized within skeletal muscle fibers, is involved in the limb regeneration. These are activated in response to injury and are involved in the repair of regenerate’s muscle tissue in the same way as this occurs in mammals. It was concluded that newt limb regeneration and muscle tissue repair in mammals share common cellular and molecular programs [117]. Differences in the sources of muscle tissue formation of the regenerating limb in the newt have also been mentioned [126,127]. The latter revealed the existence of different ways of muscle regeneration depending on the age of newts. Newt larvae use stem-like progenitors for this; mature animals use dedifferentiated mononuclear muscle progenitor cells derived from muscle fibers. As the author suggested, these differences are explained by modifications of the molecular program that regulates the death/degeneration of muscle fibers [126]. In addition, Walters et al. [128] reported the activity exhibited by senescent cells during limb regeneration in the newt. An assumption is made that the factors emitted by senescent cells activate the FGF-ERK signaling axis, thereby causing myotubules to enter the S-phase of cell cycle with production of mononuclear muscle progenitors [128]. Thus, the issue of the source of muscle tissue regeneration in the case of appendage regeneration in salamanders is still not fully addressed [129]. The presumable sources may be as follows: the migration and proliferation of mononuclear myoblasts derived from myotubules of stump muscles, the involvement of reserve cells, a combination of these mechanisms, and, as an additional possibility, the conversion of cell phenotypes already within the population of blastema cells.

In general, the issue of chondrogenesis in regeneration models was discussed earlier [130]. The major features of this tissue were noted: t is non-innervated, weakly vascularized, lacking fibroblasts, and is represented by a homogeneous cell population. However, cartilage repair has an important implication for the epimorphic regeneration of the tail with its endochondral skeleton. The conditions for the formation of cartilage, as well as muscles in the regenerate are signaling from AEC and SC cells and also the presence of progenitor cells of mesodermal origin in the blastema [79,86]. According to the classification of the stages of tail regeneration following amputation in the mature newt, based on histological characteristics [24], blastema is formed below the wound epidermis at stage II of regeneration. The blastema cells actively proliferate, and blood capillaries grow into the blastema. In the central part of the blastema, an area of greater cell density is formed: the so-called pro-cartilage condensation of cells. At stages III and IV, these cells differentiate parallel to the differentiation of regenerate’s muscles and CT. Studies using the axolotl limb regeneration model were carried out to elucidate the origin of chondrogenic cells. An attempt was made to identify blastema cell populations responsible for cartilage formation [73,131]. It was found that the axolotl’s limb blastema cells were express-paired related to homeobox1, PRRX1, and the pan-fibroblastic cell marker (FCTC), which indicated a low level of mesenchymal differentiation. With the progress of regeneration, these cells also differentiate to acquire a chondrotypic phenotype [73,131]. Note that the issues of chondrogenesis were well addressed using the model of tail regeneration in lizards [50,51,132], where, besides other findings, the similarity of the processes between lizards and salamanders was observed.

### 3.4. Pattern Formation and the Morphogenesis of the Regenerating Tail in Urodela Amphibians

It is known that appendage regeneration is nerve-dependent [133,134,135,136]. In Urodela, the tail has dorsoventral (DV) positional information that controls its regeneration following amputation. This information is based on dorsoventral differences in the SC. These differences determine the structural plan of other tissues surrounding the SC, in particular, cartilage and muscles. The morphogenetic control during tail regeneration in Urodela is also carried out in the anteroposterior (AP) direction. This patterning according to the DV and AP axes during tail regeneration has long been reported. Early, well-known experiments by Holtzer [86] with SC transplantation showed that information on the DV pattern of development is retained during the self-organization of not only the regenerating SC but also the surrounding SC tissues, muscle, and cartilage. A rotation of the SC fragment, taken from the caudal region at 180° along the DV axis, with its implantation upside down, and the amputation of the tail at the operation site gave an excellent result. As a result of such a manipulation, the SC and surrounding tissues regenerated “upside down”, i.e., with the cartilaginous tube forming dorsally of the DV axis but on the original, ventral side of the SC [86]. Thus, the SC regenerate can be considered in terms of signaling centers [123] that control the processes of regeneration and pattern formation in the tail regenerate. The key TFs responsible for patterning in embryogenesis are dorsally expressed Pax7 and Msx1/2 and the dorsolaterally expressed Pax6, which maintain a strict level of expression in both the adult native SC and the regenerating SC in the axolotl. The expression limits of these TFs control the DV orientation of tissues relative to each other in the tail regenerate of these animals [100,114].

The regulation of patterning along the AP axis is evidenced by early experiments on newts (*Notophthalmus viridescens*) with removal of blastema of the tail regenerate and its grafting into a proximal region along the longer axis of the tail [137]. The experimental data showed that the graft retains the “memory” of its original location and regenerates the amount of tail set by its original location. The patterning regeneration of SC along the AP axis has long been found in experiments [138]. The cranial displacement of the amputated SC and its replacement by paraplast leads to a loss of regenerative ability: Less than 1 mm of regenerate’s tissue is built, where muscle and cartilage tissues are not formed de novo. Simultaneously, initiation of dedifferentiation and early mitotic activity are observed. This, in turn, indicates that the initiation of proliferation and destabilization of stump cell differentiation shortly after amputation are not directly associated with SC, which, however, exerts its effect in the later stages. There is evidence that in Urodela, the neural effect for the progress of regeneration is required only from day 4 after limb amputation [139]. It has also been shown that appendage regeneration in the *Xenopus* tadpole tail and limb is nerve-dependent [140,141]. In particular, the removal of the SC leads to significant defects in the patterning and growth of the tadpole tail during regeneration [141]. Furthermore, laser ablations aimed at generating more subtle injuries within the SC at different anterio-posterior positions also result in patterning and growth defects during tail regeneration [142]. All these and other results [143,144,145] allow regarding the tail and limb blastema as a self-organizing system where cells possess positional identity along the DV and AP axes. Information on the molecular regulators of morphogenesis and patterning in tail regeneration in urodelean amphibians is provided in the following Section 4 below (Figure 3).

## 4. Molecular Regulators of Tail Regeneration in Salamanders

### 4.1. Molecular Regulatory Mechanisms Accompanying the Early Stages of Tail Regeneration in Salamanders

Above, we mentioned some molecular regulators that carry out control at various stages of tail regeneration in salamanders. This section extends this information. Signaling interactions from the moment of damage to the completion of regeneration are always changing, which makes it difficult to understand the program in its complexity and as a whole and also to identify all components of molecular regulatory networks. It was earlier found that Wnt, Fgf, Bmps, and Tgf-β pathways as well as ion channels, Shh signaling, EGF, Notch, and other signaling pathways are required during tail regeneration in salamanders [103,114,118,146,147]. The regenerating SC, in addition to the genes encoding morphogens, also displays up-regulation of genes linked to immune inflammatory response and ECM remodeling [27]. In one experiments, the morpholino-mediated knockdown of extracellular Marcks-like protein (Mlp) blocked tail regeneration. Mlp was suggested to be involved in the regulation of a wide range of processes: cell migration, secretion, proliferation, and differentiation [148]. Previously, studies using the model for limb [149] regeneration in *Ambystoma* revealed the role of macrophages in the cleaning of the damage site from dying and aging cells and in the production of pro- and anti-inflammatory molecules involved in ECM remodeling. The essential role of macrophages was emphasized by the fact that the depletion of these cells through the injection of clodronate-encapsulated liposomes blocked limb regeneration [149]. We should note here that, according to the observation [149], neutrophils and macrophages can already be detected on the first day after limb amputation. 

Below (Section 3 and Section 4), various components of molecular regulatory networks are considered that control the regeneration of the tail and the SC within it: signaling pathways, TFs, heat shock proteins (HSPs), microRNAs (miRNA), retinoid acid (RA), neurotransmitters, epigenetic regulators, etc. All the participants in the regulation were differentially activated by the timing of the process and in the space of regenerate tissues. This makes it challenging to study these participants as well as the fact that the cells of the amputated tail at the site of injury, the blastema and SC cells, are both cell sources of regeneration and sources of the signaling that formats the conditions for each of the subsequent stages of the process. Cell and tissue regeneration events either occur simultaneously or overlap with each other. For this reason, consideration of participants in the regulation of tail regeneration in a certain order corresponding to the sequence of processes is extremely conditional/provisional.

Shortly after damage, oxidative stress and reactive oxygen species (ROS) can regulate damage-induced expression of genes and proteins involved in the regulation of wound healing and regeneration [150,151,152]. The ROS signaling and also ECM remodeling and inflammation are among the first responses to tail injury [27,152,153,154,155]. Dedicated study of ROS signaling during tail regeneration following amputation was performed in mole salamanders (*Ambystoma*) [155]. By using the dye dihydroethidium (DHE), binding to superoxide anions, the authors found that ROS production occurs within the first 24 h post amputation. The activity of ROS signaling and the activity of NAPDH oxidases (NOXs) turned out to be essential for the initiation of regeneration, including post-traumatic reactions of the SC. The use of an inhibitor (DPI and VAS2870) of the activity of NOXs (that generate ROS (O_2_^-^ and H_2_O_2_)) showed that NOXs are associated with the ROS signaling pathway. In turn, the regulatory ROS signaling is associated with other signaling pathways that activate proliferation and formation of the axolotl’s tail blastema, in particular, with the Wnt/beta-catenin, Shh, and BMP signaling pathway [114,156]. Data obtained by [103] indicate that Wnt signaling can be considered a potential candidate for the redox target of NOX activity and ROS production. The role of hydrogen peroxide (H_2_O_2_) as the redox signaling molecule in the process of tail regeneration in the axolotl was studied. Exposure to H_2_O_2_ prevented the apocynin-induced growth suppression in tail blastema cells, leading to cell proliferation. H_2_O_2_ also promoted recruitment of immune cells and regulated the activation of AKT kinase and Agr2 expression during blastema formation. Additionally, on the basis of the axolotl tail regeneration model, it was found that ROS/H_2_O_2_ regulates the expression and transcriptional activity of YAP1 (transcriptional co-activator whose activity is controlled by the Hippo signaling pathway) and its target genes, *Ctgf* and *Areg* [157].

At the initial stage of regeneration, the wound epithelium, forming the AEC, plays a significant role in the molecular regulation of the process. Taking into account the data obtained using the limb regeneration model in salamanders, an opinion has been accepted that the AEC is in a reverse signaling relationship with blastema cells via growth factors, thus regulating not only their proliferative activity i but also providing blastema cells with early positional information [79] (Figure 2). As early experiments showed, the shifting of the position of the AEC laterally caused a corresponding shift in blastema cell accumulation, and transplantation of an additional AEC to the base of the blastema resulted in supernumerary limb blastema formation [158,159]. It was subsequently found that the directed migration of blastema cells is controlled by TGF-β1 through stimulation of the production of fibronectin required for this by basal AEC cells [160]. The inhibition of the TGF-β1 expression by an inhibitor of SMAD phosphorylation, SB-431542, reduced the fibronectin expression, resulting in failure of blastema formation [161].

It was also shown that the action of such signaling pathways as FGF and GGF2 and other growth factors, substances P, and transferrin have a mitogenic effect on appendage blastema cells from the wound epidermis [139]. The FGF family plays a particular role in epithelial–mesenchymal interactions between the AEC and limb blastema cells. Fgf8 and Fgf4 are expressed in the AEC formed by the wound epidermis, while Fgf-10 is synthesized in the mesoderm [139]. Fgf8 and Fgf10, in turn, are in mutual cross-regulation. An ectopic administration of FGF8 alone caused a limited regenerative response. However, the introduction of Fgf10, when interacting with Fgf8, conversely led to a more pronounced regenerative response [162]. It was also found that in the case of tail regeneration in *Ambystoma mexicanum* and the newt *Pleurodeles waltl*, regulatory interactions in the network of Fgf2+Fgf8+Bmp7 signaling pathways showed similar inductive effects. Fgf2+Fgf8+Bmp7 was involved in the tail regeneration of several tail tissues but could not organize a patterned tail [102]. An independent investigation of the FGF2 signaling pathway showed that its activity is required for SC regeneration and dedifferentiation of the stump’s mesenchymal tissues. FGF2 signaling is induced in dedifferentiated cells lining the SC canal and is observed in its cells prior to differentiation as well as in a certain subtype of neurons. Moreover, FGF2 is expressed in chondroblasts, the basal layer of the epidermis, and differentiating muscles. The introduction of ectopic FGF2 substantially enhanced the tail blastema growth [104].

Receptors mediating the FGF effect (FGFR1, FGFR2, FGFR3, and FGFR4) were studied using the model of tail regeneration after amputation in the newt [26]. The differential expression of these receptors was shown to be involved in the modulation of the FGF effect during tail and SC regeneration. It was assumed that the key ones among them are FGFR1 and FGFR4. FGFR1 is primarily associated with proliferation of progenitor cells and FGFR4 with early stages of neuronal differentiation [26].

To date, the molecular genetic signature of wound epidermal tissue cells has been studied using a model of tail amputation in transgenic larvae of *X. laevis* [76]. A cell sorting and transcriptome analysis revealed more than 8000 genes involved in the ROS, FGF, canonical and non-canonical Wnt, TGFβ, and Notch signaling pathways, which demonstrates dynamic variations in expression. 

Regeneration responses of the SC are observed simultaneously with the healing processes, AEC formation, and initiation of blastema formation during tail regeneration. It was shown that, following the tail amputation, the ependymal tube expressed the neural stem cell markers nestin and vimentin, which were undetectable in normal urodele SC [163]. As up-regulation of NSC markers has shown that ependymal cells undergo a phenotypic change. At the initial stage of SC regenerate formation, molecular signals are required for the proliferation of SC ependymoglial cells. The key, the so-called pro-regenerative, Shh, Wnt/PCP, and FGF alarm systems, were identified after both tail amputation and SC transection [26,94,104,114]. Now, it is known that the transcriptional complex AP-1 may function with tissue-specific TFs and chromatin regulators to initiate regeneration upon tissue damage [164]. To date, the AP-1 complex has been shown to play an essential role in regulating fin regeneration, *Xenopus* tail regeneration, and the axolotl’s SC regeneration [165,166,167]. The AP-1 complex directs enhanced selection to govern precise gene expression so that cells can differentiate and acquire specialized functions [168]. During the SC regeneration after transection in axolotl, the heterodimeric transcription complex AP-1 ^cFos/JunB^ and MAP kinase represent the signaling that regulates ependymoglial cells’ response to injury [97,167]. The AP-1 complex is also known to be activated in mouse and human glial cells after damage by binding to the GFAP promoter. The activation of the GFAP promoter leads to transcription activation and increased production of GFAP proteins of intermediate filaments of the cytoskeleton, thus strengthening the glial cell phenotype and making it irreversible [169]. In axolotl, on the other hand, the heterodimerization of c-Fos and c-Jun is aimed at inhibiting the GFAP expression [97,167]. Thus, the formation of non-canonical AP-1 may be a key to preventing glial scar formation and promoting a regenerative response. There is also evidence that the loss of the ability to regenerate SC in mammals is associated with the suppression of all-trans retinoic acid (atRA) via nuclear retinoic acid receptor beta (RARβ) [170,171,172,173]. In urodelean amphibians, however, the RA-associated signaling pathways, work actively during the limb and tail regeneration. It was shown that RA signaling, using the RARβ, mediates regrowth of adult SC axons towards regenerating newt limb blastemas in vitro [174]. Moreover, the expression of this receptor type is normally maintained at a low level in the SC of mature newts and is up-regulated in SC ependymoglial cells within the first 7 days after SC transection [175]. The use of LE135, an RARβ-selective antagonist, caused a significant inhibition of the SC ependymal tube outgrowth and, as a result, the regeneration of the tail in mature newt [175] and axolotl larvae [176]. The list of participants in the regulation of SC regeneration within the tail of an adult newt can also include Retinoid X receptor α [177].

As regards blastema cells, information about the regulation of their behavior from the EAC is provided above. The wound epidermis exchanges growth factors with blastema cells, thus enhancing the blastema proliferation, exhibiting phagocytic activity, and serving as a source of positional information [74]. Among the known factors affecting the behavior of limb blastema cells are factors of the FGF family, the glial growth factor (GDNF), substance P, transferrin, etc. [139]. According to [114], the Patched1 receptor expression indicated that hedgehog (Hh) signaling occurs not only within the SC but is also transmitted to the surrounding blastema of the regenerating tail. An assumption was made that the blastema cell population and growing axons of SC neurons are related by a positive feedback and are able to enhance each other’s growth (Figure 2 and Figure 3) [178].

As was found previously in the case of axolotl limb regeneration, such a well-known morphogenetic factor as all-trans retinoic acid (RA), when administered intraperitoneally, reduced the mitotic index (by 60–70%) in the blastema cell population and slowed down the growth rate and, accordingly, the entry of muscle and cartilage progenitor cells into differentiation [179]. In various urodele amphibians, RA was shown to enhance neurite outgrowth [174] and influence the specification of proximal-distal positional information [180]. Previously, it was also reported that the blastema formation not only depends on nervous system’s influence but also has a neurotropic effect itself. This phenomenon was discovered in experiments where SC fragments were co-cultured with blastema cells [178]. The blastema proved to be capable of enhancing the axon growth and contributed to an increase in their number and viability. Manifestation of the effect depended on the level of cell proliferation in the blastema. It was discovered that the factor responsible for the effect of blastema cells is the secreted neurotrophic protein molecule with a low molecular weight, the so-called mesenchyme-derived growth factor (MDGF) [180]. A detailed analysis of the molecular features of limb blastema cells in salamanders is provided in a recent comprehensive review [181], where information was summarized about numerous classes of up-regulated genes specific to appendage blastema cells in salamanders. These families constitute a range of genes associated with pluripotency, oncogenesis, and mesenchymal differentiation as well as DNA damage repair and ECM remodeling [181,182].

As was found previously, during in vitro cultivation of innervated tail regenerates of larval *Ambystoma*, the presence of wound epidermis and a fragment of the SC is sufficient for successful proliferation and differentiation, which indicates the key role of tissue interactions in this system [183]. However, the proliferation and differentiation of blastema cells is under hormonal control. In experiments with the cultivation of regenerating tail blastemas in a medium with individual hormones such as prolactin, insulin, thyroxine, hydrocortisone, and their combinations, two multidirectional hormonal effects regulating the course of regeneration were identified [184]. Prolactin and insulin promoted intensive proliferation of blastema cells but inhibited differentiation; thyroxine, on the other hand, enhanced chondrogenesis. An optimal combination of proliferation and differentiation processes, as close as possible to those observed in vivo, occurred when all four hormones were present in the medium in certain concentrations/ratios [184].

The role of ECM in the regulation of tail regeneration (and even broader, in limb regeneration in salamanders) has been discussed along with the role of the nervous system’s role [185,186]. First of all, blastema is formed through ECM degradation in the amputation site that leads to a loss of tissue organization and a release of individual cells. This, in turn, induces the loss of their phenotypic specialization and proliferation. In the study based on microarray analysis, genes were identified that showed significant changes in the transcribed mRNA abundance during the first 7 days of regeneration [27]. Initially, a range of 1036 statistically significant gene transcripts were isolated, which was then reduced to 360 gene transcripts that were used to describe predominant expression patterns and gene functions. The results showed a wide range of different molecular genetic responses to damage: However, ECM remodeling dominated this concert [27].

The spatiotemporal expression of one of the ECM components, type 12 collagen (Col type XII), in the wound epidermis, mesenchyma, and ependyma of the SC during tail regeneration was previously described from newts [187]. At 7 days post amputation, Col type XII was found in the basal layer of wound epidermis, in the ependyma, and in the mesenchyma. At 2 weeks, the epidermis and the ependyma stopped expressing Col type XII, but its expression was maintained at a high level in the mesenchyma. The gene encoding Col type XII, like other genes encoding ECM proteins, was assumed to play an important role in interactions between SC ependyma, blastema, and epidermis [188]. The protein tenascin (Tn), which is another component of ECM, and its transcripts were studied in detail using a model of tail regeneration in newts, *Pleurodeles waltl* [188]. The Tn distribution was assessed in various tissues of the regenerate and independently in SC ependymoglial cells. The results indicated a continuous and maintained Tn synthesis in the cells of the regenerating SC. A key role of Tn in the regeneration of SC axons during its recovery and during tail morphogenesis was assumed [188]. A subpopulation of cells that expressed high levels of sulfated GAGs on their cell surface was found in the axolotl skin. A suggestion was made that the GAGs produced by these cells are involved in the pattern-formation regulation during regeneration [189]. A specific, nerve-dependent spatiotemporal distribution was also described for keratins 5 and 17 using a similar model: axolotl limb regeneration [190]. Besides ECM, structural molecules and matrix metalloproteinases (MMPs) act in the stump and regenerates of appendages. Degradation of ECM by proteases after amputation of appendages disrupts contacts between ECM molecules and integrin receptors. This, in turn, leads to variation in the shape of cells and reorganization of their cytoskeleton. Cytoskeletal rearrangements cause the activation of signaling pathways of transduction, which induces up-regulation of enzymes that dismantle the phenotype-specific internal structure of the cells [79]. The role of the 90-kDa gelatinase was studied earlier on a model of axolotl limb regeneration [191]. These proteins appeared to be active during ECM remodeling in the distal region of the stump and 4 mm proximally of the regenerate, where their function was aimed at releasing progenitor cells and inducing their dedifferentiation to form blastema. Subsequently, the intensity of expression decreased and reached the baseline level observed prior to the operation. It was suggested that 90 kDa gelatinase/collagenase may play a role in the initiation and rapid growth phase of regeneration and wound healing in the axolotl [191]. Mmp9 expression was studied using a model of axolotl limb regeneration [192]. This expression was detected in the wound epithelium as early as at 2 h post amputation and reached a peak at 14 h. The authors suggested that Mmp9 facilitates the epidermal–mesenchymal interactions required for successful regeneration. Besides the above-mentioned studies, expression of many other metalloproteases (Mmp1, Mmp2, Mmp3, Mmp8, MMmp10, Mmp12, Mmp13, and Mmp19) is also reported in a number of other publications [193,194,195,196]. It is emphasized that the chemical inhibition of these metalloproteases greatly disrupts regeneration, which indicates a key implication of ECM remodeling in regeneration.

A number of studies revealed a relationship between the expression of ECM components, TFs, and growth factors. Thus, Satoh et al. [196], based on a model of axolotl limb regeneration, studied the expression of an orthologue of Twist (AmTwist), a basic helix-loop-helix TF involved in dermal tissue regeneration. They found that AmTwist is inhibited by signals from the nerve during the early stages when dermal fibroblasts dedifferentiate to form blastema cells. However, a relationship of AmTwist expression with an ECM protein, type I collagen, in the proximal region of the blastema was recorded. Exogenous BMP2 led to an increase in AmTwist expression as an endogenous regulator of AmTwist expression and dermis regeneration. This study also highlights another function of ECM components: not only involvement in the remodeling of matrix and cell surface but also in the action of regulatory networks [197]. Recently published evidence [198] shows that FGF signaling serves as an inducer of collagen regeneration and the formation of collagen fibrils during skin wound healing in axolotls. Information was also presented on the regeneration of knee cartilage in newts [199]. This model also provides a view of the ECM-dependent mechanisms of cartilage repair in tail regeneration. After applying damage of two types (surgical removal and collagenase treatment), the authors [199] identified molecular regulators of cartilage repair. To identify relevant candidate genes and determine their impact on regeneration, they used cDNA microarray analysis, real-time quantitative PCR, immunohistochemistry, and functional assays, which made it possible to identify groups of up-regulated genes in the cartilage regeneration process. These include genes encoding ECM components, mediators of cell–ECM interactions, tissue remodeling, signaling molecules, homeostasis, etc. Of particular attention was the expression of tenascin C (Tn-C), a matricellular protein up-regulated during cartilage repair in both damage models [188].

Figure 4 schematically shows a set of events of tail and SC regeneration prior morphogenesis/patterning and key molecular participants of these steps regulation (Figure 4).

### 4.2. Molecular Regulators of Morphogenesis/Pattern Formation in Tail Regeneration

Currently, it is obvious that a regulatory signaling pathways capable of coordinating regeneration processes is involved in the control of tail regeneration, morphogenesis regulation, and pattern formation. As can be seen in the previous sections, the regulatory machinery acting at the early stages of tail and SC regeneration determines the subsequent and final stages of regeneration: the histo- and morphogenesis of the regenerate. It is known that the dorsoventral (DV) and anteroposterior (AP) polarities, characteristic of tail tissues and the organ in general, are maintained during the process of tail regeneration in Urodela [114]. The length of the tail regenerate is directly proportional to the length of the amputated area, and the number of myomeres and vertebrae in the regenerate is proportional to their number in the amputated part of the tail [137]. To date, the major molecular genetic mechanisms and programs responsible for reproducing the pattern formation in both DV and AP directions during tail regeneration in urodelean amphibians have been described (Figure 3). This issue is preliminarily discussed above (Section 3.1 and Section 4).

Maintaining the polarity along the DV axis is extremely crucial for the correct arrangement of structures and the formation of correct neural links in the regenerate that provide restoration of the tail’s function. Numerous earlier studies (reviewed by [200,201,202]) provided information on the existence of subdomains characterized by the expression of TFs, containing homeodomain and pair-box domain proteins, in the neural tube during development in vertebrates. It was found that the dorsal side is characterized by the expression of *Msx1*,*2* genes, the dorsolateral side by *Pax7*, and the lateral side by *Pax6*, whereas ventral domain cells are characterized by the expression of the *Nkx6.1* and *Nkx2.2* genes. It was also recorded that the size and position of these domains are controlled by counter-directed gradients of morphogens: the ventralizing gradient of Shh and the dorsalizing BMP. In development, the signaling factor Shh is expressed in the notochord and the floor plate, inducing the concentration-dependent differentiation of cell types in the ventral region of the neural tube [203,204,205]. The expression of type I homeotic genes (*Pax7*, *Irx3*, *Dbx1*, *Dbx2*, and *Pax6*) is inhibited in the presence of Shh, while the expression of class II homeotic genes (*Nfx6.1* and *Nkx2.2*) is activated. Low concentrations of Shh block the expression of *Msx1* and *Pax7* but increase the expression of *Pax6* in cells of the lateral region of the neural tube. A study of *Patched1* receptor expression detected the work of Shh signaling both inside the SC and in blastema cells [114]. Blocking the action of the Shh signaling pathway by exposure to cyclopamine demonstrated that both the SC patterning and the tail regeneration in general depend on it. Shh induces the cartilage formation from the ventral side of the ependymal tube and also regulates the myogenic cell differentiation by activating the Myf5 expression [114]. The proliferation of blastema cells and the expression of *Sox9* in cells of its ventral region depend on the Shh signaling pathway. A conclusion was also drawn that one of the regulators of cartilage induction from the SC side in the axolotl’s regenerating tail is also the Shh signaling [114]. In the SC development, the Shh signal transmission is mediated by genes as transcriptional mediators that encode Zinc-finger proteins of the Gli family, namely *Gli1*, *Gli2*, and *Gli3*, regulating such TFs as AxFKH1 and AxFKH2. Studies indicate that additional signals may provide positional information in parallel to Shh to specify the neuronal fate in SC tissue [206]. The authors suggested that various upstream patterning signals may be integrated by the Gli proteins to direct a coherent program of neurogenesis [206]. It should be emphasized here that the genes of the key Shh signaling pathway play an important role not only in the structuring of the SC but also in the regulation of cell proliferation, organization of patterning, and specification of cell types. This, in turn, provides the coordinated development of the SC and surrounding mesodermal structures. Defects in all tissues extend beyond the normal sites of Shh transcription, thus confirming the proposed role of Shh proteins as an extracellular signal required for the tissue-organizing properties of several vertebrate patterning centers [207].

The gradient of Shh is set against the gradient of the BMP morphogen, a family of regulatory molecules synthesized in the ectoderm and the spinal cord roof. BMP4 and BMP7 activate the expression of the *Msx1*, *Pax7*, and *Pax6* genes in the dorsal and lateral parts of the neural tube. These domains are present in both the developing SC and the ependymal tube of the regenerate. In development, the dorsoventral organization of the neural tube is determined by signals from surrounding mesodermal tissues [208] and, in regeneration, by the positional information contained in the adult SC [114]. BMP signaling is a key component in appendage development and regeneration in amphibians [209,210,211]. In clawed frog (*Xenopus*) larva, the ability to regenerate the tail can recover at a non-regenerative competent stage by activation of BMP signaling and vice versa: If the BMP pathway is inhibited at a regeneration permissive stage, then the regeneration can be inhibited [209].

The regulators that provide the determination of progenitor cells and the fate of SC neurons include the genes encoding such TFs as Math1, Mash1, and neurogenin, which form a complex system of mutual inhibition [212]. In the newt, the homeobox gene *PwDlx3* is expressed in the epidermis, in cells associated with muscle masses, and in the ventrolateral parts of the ependymal tube. The authors suggest this gene might be expressed in cells that have some neural crest cell potentialities [213]. Thus, we see that the regeneration of the SC, tail, and limbs is associated with the activity of signaling pathways and the re-activation of genes involved in tissue development and in the formation of patterning in the structures formed de novo.

When considering the patterning of the regenerating tail along the AP axis, as in the case of the formation of structures along the DV axis, one should take into account the normal nervous system development. The family of genes encoding the components of the Wnt signaling pathway has an important implication for the formation of the primary and secondary axes of the embryo [214]. The use of the model of tail regeneration in *Xenopus* tadpoles showed that Wnt acts upstream of other developmental signaling pathways such as BMP, Notch, and FGF in establishing muscle and neural cell fates in regeneration [209,215]. In caudate amphibians, many genes of the Wnt family work at a low level in the intact SC but, after an injury-induced stress, are eventually up-regulated [216]. During tail regeneration in newts (*Pl. waltl*), the work of the following genes intensifies: *Pwnt-5a*, *Pwnt-5b* [217], and *Pwnt-10a* [217]. These genes are expressed at a high level in the blastema, with a maximum expression in the most caudally located region. The level of *Pwnt-7a* expression also increases in the epidermis of the tail regenerate [218].

The interaction of two key signaling systems, Shh and Wnt, was studied during the regeneration of the tail and SC after amputation in newts [118]. Using q-PCR of mRNA, transcripts of various components of these signaling pathways were found that are activated in the case of tail damage: shh, ihh, ptc-1, wnt-3a, b-catenin, axin2, frizzled (frzd)-1, and frzd- 2. A long-term pharmacological inhibition of Shh signaling led to the development of an atypical spike-like regenerate with no visible tissue organization in it [118]. The inhibition of Shh-mediated patterning takes place as early as at the first regeneration stages, at the stage of blastema cell proliferation. The inhibition of blastema cell proliferation was shown using the BrdU and PCNA markers. Based on observations of [114,118], one may conclude that the Shh signaling, modulating the expression level, like Wnt, plays an essential role in the regulation of cell proliferation throughout the regeneration period, and Shh morphogens are necessary for transitive patterning at the onset of this process. The inhibition of Wnt as well as Shh signaling causes disruption of the regeneration process, while the activation leads to its promotion. 

According to the data of [219,220,221], the structural plan along the AP axis is determined by expression of the Hox-a and Hox-b cluster homeobox genes. In the newt’s tail regeneration, many genes of this family resume their action and are expressed at a much higher level than in mature tissue: *Hoxb13*, *Hoxc10* [144], *Hoxc13*, and *Hoxc12* [145]. As the studies on the expression of this group of genes in sexually mature newts showed, *Hoxa9*, *Hoxc10*, *Hoxc12*, and *Hoxc13* are expressed at various axial levels of the SC and are capable of re-expression during tail regeneration [145]. Re-expression of the *Nkx3.3* and *Nkx3.2* genes responsible for the AP gradient was observed in the wound epidermis, neural tube, regenerating muscles, cartilage, and spinal ganglia of the newt *Pl. waltl*. It was emphasized that these genes play a crucial role in regulating the regeneration of skeletal structures [222]. The regulatory role of the TF Hif1α, a stress-induced TF, and its relationship with the Wnt signaling pathway was assessed using the model of tail regeneration in *Xenopus tropicalis* [223]. It was found that Hif1α is required for the regeneration of differentiated axial tissues, including axons and muscle. Using RNA sequencing, the authors discovered that Hif1α and Wnt converge on a broad set of genes required for posterior specification and differentiation, including the posterior *hox* [223].

During regeneration of the salamander’s appendages, blastema cells are capable of the interpretation of proximal-most positional identity in the stump to reproduce the missing parts faithfully [224,225]. Earlier classical experiments showed the ability of RA to proximalize blastema positional values [226].

RA and the RA receptors are regulatory molecules for both DV and AP patterns and have been partly discussed above. The role of RXRα and RARβ, members of the family of retinoid receptors, was identified [175,177]. It was found that the use of the RXR agonist, SR11237, results in the down-regulation of RXRα and, on the other hand, up-regulation of RARβ, which leads to the inhibition of tail and SC regeneration in the newt. The results showed the existence of a regulatory feedback loop between two subtypes of RA receptors as a mechanism regulating the regeneration of tail and SC tissues in regeneration-competent newts.

Meis homeobox genes are involved in the RA-dependent specification of proximal cell identity during limb development in the axolotl [227]. To understand the molecular basis for specifying proximal positional identities during regeneration, Mercader et al. [227] isolated the axolotl Meis homeobox family. During limb regeneration, Meis overexpression was found to relocate distal blastema cells to more proximal locations, whereas Meis knockdown inhibited RA proximalization of limb blastema. A conclusion was drawn that the Meis genes are thus crucial targets of RA proximalizing activity on blastema cells [228].

### 4.3. Epigenetic Factors of Regulatory Control of Tail and SC Regeneration in Urodela

Epigenetic factors that carry out precise regulation of gene expression undoubtedly play an important role in the processes of re-patterning regulation during regeneration as well as patterning in development [164,229]. The major epigenetic mechanisms include DNA methylation, histone modifications, activity of small and non-coding RNAs, and chromatin reorganization. Studies of changes in the epigenetic landscape during the epimorphic regeneration of appendages in amphibians are rare.

It was found that during the tail and SC regeneration in Urodela, miRNAs become participants in the regulation of cell behavior [10,230,231]. The generally recognized function of miRNA is to be involved in RNA silencing and post-transcriptional regulation of gene expression [232]. The role of AP1(cFos/JunB) in combination with miR-200a was identified as a regulator of the suppression of reactive gliosis and pro-regenerative responses of glial cells during SC regeneration in the axolotl [167]. In [10], the function of miR-200a in maintaining the “stemness” traits in NSCs after SC transection in the axolotl was identified by inhibiting in them the expression of a marker of mesodermal cells, the *brachyury* gene, which is a founder member of the TFs family that shares the so-called T-box-a 200 amino acid DNA-binding domain [10].

Besides miRNA-200, another miRNA (miRNA-196) was earlier identified as an essential regulator of tail regeneration, controlling blastemal cell positional identity and size of the regenerate [230]. miRNA-196 is able to regulate the rates of cell proliferation and early blastema growth. In a study of the axolotl’s tail regeneration, it was found that Pax7, BMP4, and Msx1 are up-regulated in the animals treated with a miRNA-196 inhibitor. This suggested that miRNA-196 is acting directly on upstream of one or more of these genes. A detailed investigation into the role of miRNA-196 showed that it acts directly on Pax7 to down-regulate the Pax7 protein level in cells in a 500 μm zone anterior to the plane of amputation. The authors identified the binding site for miRNA-196 in the 3′UTR of the axolotl Pax7, suggesting that protein levels are affected by increasing or decreasing the levels of miRNA-196 and Pax7. This, in turn, affects blastemal cell division, giving rise to the short tail phenotype in inhibitor-treated axolotls. These data also suggested that Pax7 acts in a feedback loop with BMP4 and Msx1 to regulate both blastema proliferation and patterning during regeneration [230].

In a wide study performed by [233], the authors, based on deep sequencing combined with qRT-PCR, carried out a comprehensive identification of miRNAs involved in regulating regeneration in the axolotl. Specifically, among the miRNAs that the authors found to be expressed in axolotl tissues, the 4564 microRNA families were identified, which are known to be widely conserved among vertebrates. These findings support the hypothesis that miRNAs play key roles in managing the precise spatial and temporal patterns of gene expression, which ensures the correct regeneration of missing appendages [233]. It is also known that during fin regeneration in *Danio rerio*, miRNA-133 serves as a negative regulator inhibiting the work of genes associated with cell proliferation, in particular, the *mps1* gene. However, during regeneration, the miRNA-133 pool is depleted. With the experimental introduction of ectopic miR-133, the regeneration rate slowed down, and conversely, inhibition of endogenous miRNA-133 increased the proliferation of blastema cells, thus contributing to an increase of the regenerate in size [234].

Tissue regeneration is associated with complex changes in gene expression and post-translational modifications of proteins, including the TFs and histones that comprise chromatin. The involvement of one of the components of epigenetic regulation, histone deacetylase (HDAC), was studied using an inhibitor of valproic acid (VPA, a well-known teratogen) in cases of tail and limb regeneration in the larvae of *Xenopus laevis* and *Ambystoma mexicanum* [235]. The 7-fold inhibition of HDAC activity in that experiment led to suppression of appendage regeneration in both animals. Another HDAC inhibitor, sodium butyrate, was also shown to inhibit tail regeneration. The authors came to a conclusion that histone deacetylation is specifically required for early events in appendage regeneration in amphibians and suggested that it may act as a switch to trigger re-expression of some of developmental genes [235]. Subsequently, in [236], 172 compounds were tested that had been designed to target epigenetic mechanisms in an *Ambystoma mexicanum* embryo tail regeneration assay. A relatively large number of compounds [55] inhibited tail regeneration, including 18 histone deacetylase inhibitors (HDACi). Among them, special attention was paid to romidepsin, which inhibited the regeneration process under 7-day exposure. A microarray analysis showed that romidepsin altered early transcriptional responses (at 3 and 6 h post amputation). Targeting genes that are implicated in tumor cell death as well as the genes that function in the regulation of transcription, cell differentiation, cell proliferation, pattern specification, and tissue morphogenesis were among them. A conclusion was drawn from the results obtained that HDAC activity is required at the time of tail amputation to regulate the initial transcriptional responses to injury and regeneration [236]. A modification of histones was investigated using immunohistochemistry methods on a model of tail regeneration after amputation in mature newts, *Pl. waltl* [229]. Histone H3 is one of the most extensively modified histones among five primary histone proteins. Cell proliferation and acetylation/methylation of H3 histones were assessed at 3 days post operation. It was found that the amputation provoked acetylation of H3K9, H3K14, and H3K27 but did not significantly change the methylation of H3K27 around the residual stump in the region of high-proliferative activity of cells. The authors concluded that, despite the lack of direct evidence, epigenetic modifications are likely to be involved in regenerating the amputated tail in the newt. It should also be noted that histone acetylation was previously reported to increase in the tail of an *X. laevis* tadpole within 24 h after its amputation [237].

The following are the main patterns of the tail and spinal cord regeneration, which we have emphasized in this review (Figure 5).

### 4.4. Morphogenesis, Known Changes/Disruptions, and Their Presumptive Molecular Regulators

#### 4.4.1. Known Morphological Changes Appearing Evident at the Late Stages of Tail Regeneration in Urodelean Amphibians 

The mutual influence of tissue mechanics and the molecular regulatory signaling accompanies histogenesis and patterning in a developing organ. This is true for regeneration as well as for normal tissue development [32,33]. The intrinsic regulatory program of morphogenesis works in a spatiotemporal order, resulting in the formation of the organ strictly corresponding to the functions that it will perform. However, the morphogenetic program can be modified, causing external changes determined by both growth rate and shape deviations. The question as to how this happens becomes especially relevant and requires research where the musculoskeletal systems of animals is considered. In regard to the tail, the major needs for interdisciplinary scientific research are emphasized, and the great contribution that studies make in at least five fields is discussed: (1) evolution and development; (2) regeneration; (3) functional morphology; (4) sensory/motor control; and (5) computer and physical modeling [54]. The issue of modulations/disruptions in the regenerating tail’s morphogenesis has not been sufficiently studied to date. The studies mentioned below are only some of the few attempts to address it. Nevertheless, this regeneration model provides the widest range of opportunities for studying a variety of internal and external causes of growth rate modulations, the shape of the regenerating structure, and the cellular and molecular mechanisms of their occurrence.

The stages of tail regeneration appear to be conserved among salamander species that represent different tail morphologies, e.g., between terrestrial mature newts that have relatively thin tails and aquatic *Ambystoma mexicanum* that have wide, keel-shaped tailfins adapted for swimming. Presumably, regeneration is also conserved between male and female salamanders of species that exhibit sexually dimorphic tail morphologies, with, however, the effect of genetic sex on tail regeneration remaining unknown. A series of papers by [56,238,239] raised the question of genetic causes of morphological modulations during tail regeneration in salamanders. These studies were based, in particular, on crossing of *A. mexicanum* with its closely related species and a linkage analysis of localization of genetic factors. In a study by [240], this approach made it possible to judge the innate genetic factors associated with tail shape variations. Terrestrial forms of post-metamorphic hybrids *Ambystoma mexicanum* × *A. andersoni* were obtained. In the hybrid animals, two amputations of the distal part of the tail were performed with further observation of the regenerate’s growth. The genotype of each of these animals was studied on the basis of 187 molecular markers detected in DNA isolated from them. As a result, a strong correlation between the shape of the tail regenerates and the sex of the animals was found after both amputations. Regeneration of longer and wider tail regenerates was observed in 66–68% of cases in males compared to females, which had shorter and narrower tails. This phenomenon was explained by the genetic factor *ambysex*, the sex-determining locus on LG 9. Thus, the results of the study [240] showed that mainly *ambysex* and only a very little effect of other quantitative trait loci can explain variations in the morphology of tail regenerates in the hybrids. Thus, a genetic factor was identified that determines the differences in the regenerating tail shape in *Ambystoma* salamanders.

Furthermore, there are a few more judgments as regards the rate of regeneration and variations in the shape of the de novo forming tail. It is suggested that endocrine signals are likely to be mediators of these effects [241]. Endocrine signals acting directly on receptors expressed in the tissue or via neuroendocrine pathways can affect the regeneration by regulating the immune response to injury and allocation of energetic resources or by enhancing or inhibiting proliferation and differentiation pathways involved in regeneration. The issues of endocrine regulation of regeneration are discussed in the review studies [241,242].

The effects of various external factors on the tail regeneration and the level of recovery of its functions are considered in the studies on semiaquatic plethodontid salamanders [243,244]. It was claimed that the tail regeneration rate is dependent on the temperature, body size, and amount of tail length lost. In particular, it was shown that the time required for amount of tail length regeneration (about 63–143 days) increased significantly with body size [243]. To examine variations among seasons and environments in the cost of tail autotomy, reference [244] tested the effect of temperature, photoperiod, and feeding on tail-length re-growth in *Desmognathus conanti.* It was found that a low temperature (10 °C) had a large, negative effect, but the photoperiod did not. The pronounced thermal effect resulted from a combination of delayed initiation of tail length re-growth and reduced regeneration rate thereafter at a low temperature.

#### 4.4.2. The Role of Heat-Shock Proteins in Appendages’ Regeneration in Anamniotes 

The role of heat-shock proteins (HSPs) in relationship with the effects of various factors in regeneration models has not been sufficiently studied. However, HSPs of 40, 60, 70, and 90 kDa have been shown to be synthesized in response to damage and to be present in regenerate cells in various animals. In planarians, HSPs are essential to maintain a pool of neoblasts that constitute a cell source of reparative regeneration. In other cells, their synthesis is mainly triggered in response to stress, whereas in neoblasts, it is carried out constitutively. Knockdown of HSPs caused inhibition of both growth and regeneration in planarians (see the review [245]). As was shown using a *Danio rerio* model, Hsp70 is not only the first to be induced by damage to the optic nerve but is also required for its successful regeneration [246]. The introduction of an inhibitor (HSP inhibitor I) slowed down the axon growth and eventually led to disruptions in the optomotor behavior of the fish [246]. In the case of the caudal fin regeneration in the same animal model, Hsp60 expression was observed in cells of the regenerate blastema. Mutation in the Hsp60 gene led to mitochondrial defects and apoptosis of blastema cells, disruption of blastema growth, and inhibition of fin regeneration [247]. Using transgenic *X. laevis*, in which the BMP signaling pathway and, accordingly, the hind limb regeneration after amputation were experimentally blocked, it was found that the BMP downstream genes necessary for regeneration include *Hsp60* [248]. In this study, based on the Affymetrix Gene Chip analysis, the authors identified genes linked to regenerative success downstream of BMP signaling. The Gene Ontology analysis showed that the genes involved in embryonic development and growth are significantly over-represented in regenerating early hindlimb buds and that successful regeneration correlates first with the induction of stress response pathways [248].

A limb amputation in the newt *N. viridescens* caused the synthesis of HSP90, HSP70, HSP68, and HSP30 in the injury zone within the first hours post operation [249,250]. As reported in the study [250], the expression of a number of stress-induced proteins, including HSPs, significantly increased within a few hours after limb amputation in newts of this species. When non-operated animals were kept at a temperature of 37 °C, there was also an increase in the expression of HSPs, of which some were similar in molecular weight with the detected stress-induced ones. It is assumed that the spectra of stress- and heat-shock-induced HSPs may differ. It is also known that the *Hsp70* gene is activated during the limb regeneration not only of newts but also of the axolotl *Ambystoma mexicanum* [251]. As was found, the level of Hsp70 expression in intact tissues is low but increases significantly at 24 h after limb amputation, and after reaching a maximum, it persists in blastema cells [251].

In a recent study, based on a model of retinal regeneration after detachment in the newt *Pl. waltl* and RNA sequencing, the up-regulation of classic early response genes, including HSPs, chaperones, and co-chaperones, was detected among 1019 transcripts [252]. Thus, the available data obtained from various animal models as well as regeneration models indicate the promising perspectives of studying the expression and role of HSPs in appendage regeneration due to their undoubted involvement in this process. Note also that stress proteins are used as biomarkers in the assessment of various biological effects of environmental agents on the organisms at molecular levels. The levels of heat-shock proteins could be especially significant in the environmental adaptation for many aquatic organisms [253].

#### 4.4.3. The Effect of Gravity Doze on the Morphology of Regenerating Tail in the Newt

For a long time, newts have been regarded as a promising and convenient animal model for space biology studies [254,255,256]. These animals were used, for the first time, to study the regeneration of eye tissues and appendages after exposure to space conditions (review by [257,258]). In the experiments aboard the Foton-M2 and -M3 unmanned spacecrafts [259,260], we compared the morphologies of the regenerating tail in newts *Pl. waltl* (normally having a semi-aquatic lifestyle with neutral buoyancy) exposed to the conditions of spaceflight (micro-g) and in animals of 1 g control that were kept on a solid wet substrate (Figure 6 and Figure 7).

In the latter case, there was a striking morphogenetic effect. The 1 g conditions caused uniform and stable morphological alterations in the tail during its regeneration post amputation. The effect of 1g was manifested as a deviation of all tissues of the forming tail, which started curving downwards. The tail regenerate had a hook-like shape instead of the usual lanceolate shape observed in animals kept in an aquatic environment in a tank or in a flight. Interestingly, in an experiment under 2 g hypergravity simulated with a centrifuge, we observed the same picture of an altered shape of tail regenerates: They became curved downwards [44]. The observed morphogenetic effect of a non-specific physical factor applied to a complex structure of tail regenerate in a mature animal has appeared to be a rare phenomenon with an unknown molecular basis.

To test the possible role of HSPs in the identified changes in the tail regenerate shape, we used a morphometrical analysis of tail regenerate shape in a tank (control), on a substrate (1 g, a relative hypergravity for newts), and in a tank under weekly exposure to heat shock [43,45]. Thus, we studied the gene expression and protein localization of HSPs with a molecular weight of 70 and 90 kDa. The weekly exposure of the newts’ regenerating tails to heat shock in otherwise normal conditions caused the development of curved tails (both upwards and downwards), suggesting that similar mechanisms are activated in morphogenesis altered by both 1 g and 2 g as well as heat shock. The heat-shock protein inhibitor KNK437 did not affect the tail shape during normal regeneration but prevented the formation of a curved tail in appropriate conditions. In addition, it was shown that the HSP70 and HSP90 proteins are present in the muscles and CT of intact tails as well as regenerates but appear only in the epidermis in hypergravity-altered regenerates and heat-exposed tails. Based on these data, we hypothesize that different external factors (e.g., 1 g gravity, 2 g hypergravity, and heat shock) induce signals that are received, analyzed, and transmitted further to affect morphogenesis mechanisms similar to those that utilize a set of HSP in epidermal cells [45]. Deciphering the molecular mechanisms underlying the modulations of regenerating tail morphogenesis is fundamental for developmental and gravitational biology and needs further research.

## 5. Conclusions

Urodelean amphibians have the ability to regenerate the tail and the spinal cord (SC) in it, which is a rare example of the regeneration abilities of vertebrates. The capability of forming a functionally complete organ de novo is assumed to be predetermined mainly by the pedomorphic state of these animals. The de novo formation of the tail after amputation passes three main stages characteristic of the epimorphic regeneration: (1) wound healing and activation and mobilization of cell sources located in the stump tissues; (2) accumulation of blastema cells, formation of apical epithelial cap (AEC), and regrowth of SC; and (3) differentiation of cell-specific phenotypes, histogenesis, morphogenesis, and patterning. All these processes are interdependent, strongly coordinated, and aimed at the natural restoration of the tail as a whole, fully functioning organ.

Each of the stages is characterized by the molecular genetics features of both the cells involved in the process and the regulatory networks that control all aspects of the behavior of these cells. The regulatory networks consist of both organism-wide factors and short-range signaling pathways, RA, HSPs, ECM-remodulating regulatory components, and, finally, TFs and epigenetic factors regulating gene expression.

The AEC and the SC as well as blastema cells play a crucial role in regulating the regeneration and are considered in terms of centers of regulatory signaling. The mutual molecular effects of the AEC, SC, and blastema are carried out by the feedback loop principle. The features of regulation of genetic expression, controlling signaling pathways, and their molecular components have been studied for these cell populations.

In histogenetic and morphogenetic events and patterning during the regeneration of the tail and the SC in it, Urodela, to a certain extent, recruit the molecular regulatory mechanisms that act in development during the formation of the DV and AP axes of the developing SC and adjacent tissues and perform “marking” of their relative orientation, size, and structure.

The effect of environmental factors in changing the shape and parameters of the regenerating tail has been poorly studied. The situation has arisen because environmental effects have been commonly beyond researchers’ main attention, focused on understanding the basic cellular and molecular patterns of the tail and SC regeneration. Our preliminary data obtained in studies on gravity-dependent changes of regenerating tail morphology suggest the possible implications that differential changes in the expression of heat-shock proteins have for tail regenerate tissues. 

In addition to these special issues, extensive information obtained using the well-known classical model of epimorphic regeneration—regeneration of the tail and its SC in urodelean amphibians—can contribute to addressing a wide range of issues in regenerative biology and medicine. These issues cover the following: (1) the issue of approaches to the induction and stimulation of SC regeneration in mammals; (2) the mechanisms of morphogenesis of tissues and organs in ontogenetic development and in the mature organism; and (3) the mechanisms of the effect of various external factors on cell populations and on the course of appendage regeneration in vertebrates.

## Figures and Tables

**Figure 1 life-14-00594-f001:**
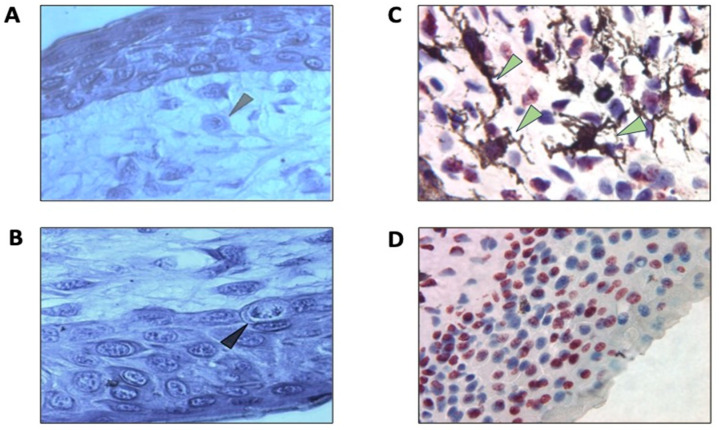
Morphological pictures of blastemal cells (**A**,**C**) and apical integumentary epidermis (**B**,**D**) of tails regenerating after amputation in the newts *Pleurodeles waltl* (10 days p/amp). (**A**–**C**) Magnification: 400×. Black arrows show mitotic figures; (**C**,**D**) plenty of BrdU-labeled cells of brown color among blastemal and epidermal cells. Green arrows, melanocytes among blastemal cell mass. (**D**) Magnification: 200×.

**Figure 2 life-14-00594-f002:**
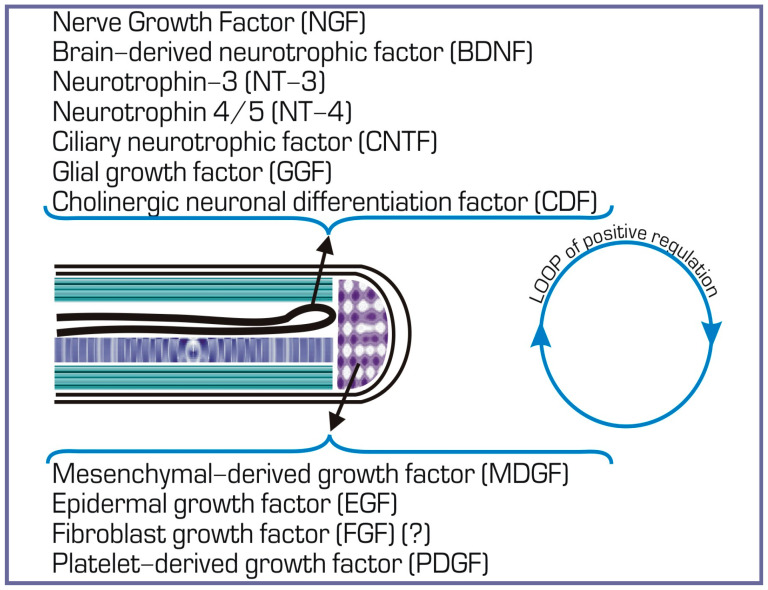
Neurotrophic and growth-support factors in the outgrowing spinal cord, proliferating blastemal cells, and cells of the apical epidermal cap in the regenerating tail of urodelean amphibians (Scheme).

**Figure 3 life-14-00594-f003:**
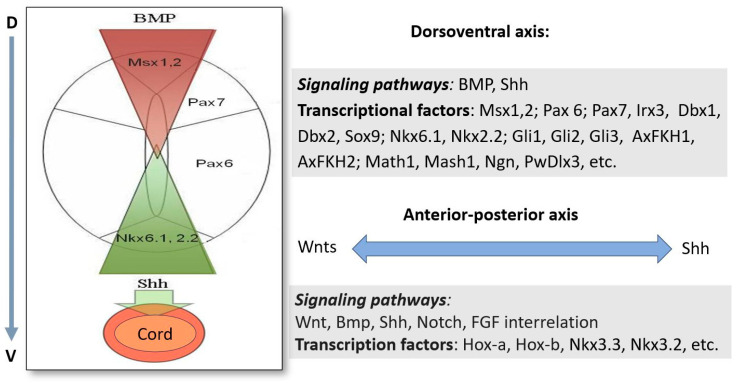
Key molecular players in SC and adjacent tissue patterning in tail development and regeneration along D/V and A/P axis. Schematic representation. The detailed description is in the text.

**Figure 4 life-14-00594-f004:**
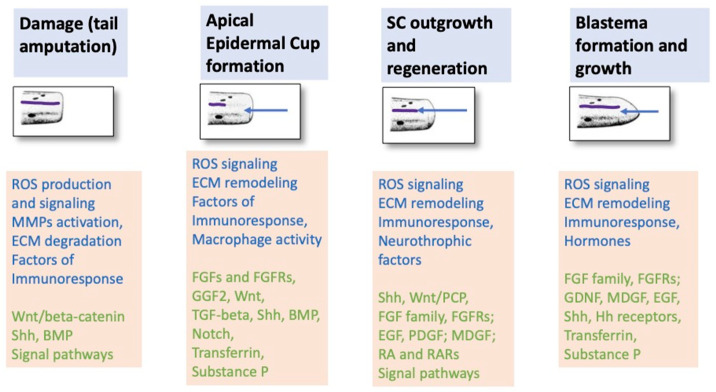
Early steps of the tail and SC regeneration in urodelean amphibians and the main secretory, short-distance molecular players participating in their regulation. Cell and tissue regeneration events either occur simultaneously or overlap with each other. For this reason, consideration of participants in the regulation of tail regeneration steps is conditional. More detailed description is in the text.

**Figure 5 life-14-00594-f005:**
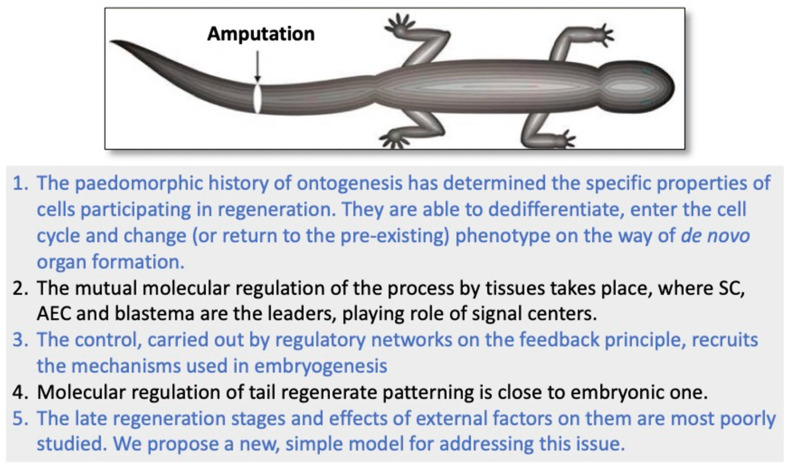
The main patterns of the process of tail and spinal cord regeneration in Urodelean amphibians, which we did try to highlight and emphasize in the paper.

**Figure 6 life-14-00594-f006:**
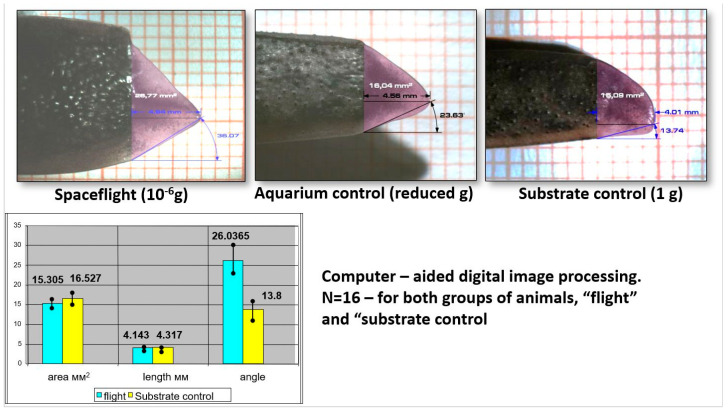
Comparison of the parameters (size and bending angle) of tail regenerates that were formed in Foton-M3 spaceflight under microgravity conditions and in the desktop control on a solid substrate in the lab.

**Figure 7 life-14-00594-f007:**
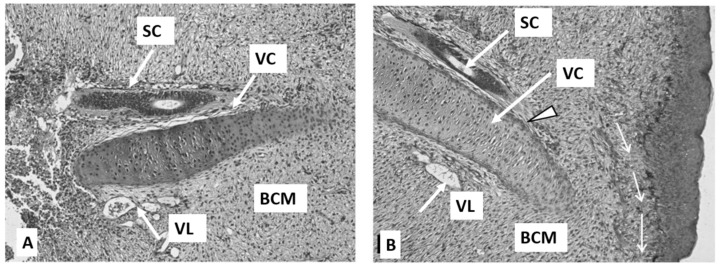
Morphological pictures of the apical region of tail regenerates after amputation in the newt *Pl. waltl.* (**A**) Operated newts kept in a deep aquarium; (**B**) operated newts kept on a solid substrate. Magnification: 40×. SC, spinal cord; VC, vertebrate column; VL, vessel lumen; BCM, blastemal cell mass. Thin arrows show the bending direction; thick arrows show the place of bending.

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
