# Peer review of "Tail and Spinal Cord Regeneration in Urodelean Amphibians"

_life, 2024, doi:10.3390/life14050594_

Round 1

Reviewer 1 Report

Comments and Suggestions for Authors

General

The present MS deals with a CLASSICAL although less studiers aspect of amphibian regeneration, in the tail. To write a MS on such an overexploited topic of the regenerating literature is demanding these modern days, in consideration to the numerous extensive papers published in the last 50 years, at least. A new MS should bring news that address the topic to some better understanding, especially in relation to human regeneration. The MS is contains an original synthetic essay of previous information on the topic but should be better organized in a more synthetic form, and introducing a number of sub-headings, and also some drawings/tables that visually express/summarize the author’s ideas and descriptions on the topic. Also, giving the large and often repetitive literature on the topic, the present MS should be considerably shortened and principally limited to the NEW and INTERESTING IDEAS reported in the present MS. Also, some important and key papers on the topic are neglected, and should be instead considered here (see below). My suggestions only aim to produce a LONG paper of interest for readers, also on behalf of the present Authors efforts in making it.

 Specific

 -line 31:….are capable of full regeneration….. : AT LEAST 2 studies (Stensaas L LJ. Regeneration of the spinal cord……….In: Spinal cord reconstruction….CC Kao et al (Eds) Raven press. AND in  Davis et al., Time course of salamander…..J Comp Neurol 293, 208-,1990 ) show that SC regeneration in salamanders IS NOT 100% like the original SC anatomically and functionally, although is very close to “full”. This is because-even in the Best regenerators- all developmental steps leading to SC embryogenesis cannot be repeated in adult urodeles.

-lines 53-63: these ideas are outdated and repeats the usual concepts of many other papers. No tail SC is present in mammals, and amphibian models are therefore useful mainly for amphibians and for getting new data on THEIR specific regeneration, an important but limiting point to be considered. The extrapolation of these infos to other species, especially amniotes, is problematic and requires enormous knowledge on gene pathways far away to be known.

-lines 78-79: digit regeneration is regengrow (healing+somatic growth), ear hole is extensive wound healing, horn regeneration is bony regeneration mainly. It is customarily called “regeneration” but distinctions should be made considering extensive tissue healing (not regeneration that implies re-patterning etc.) and regengrow.

-Fig. 1 A appears as a TOO HIGH cell concentration and un-usual cell aspect for an amphibian blastema. Cells should be more loosely distributed (see 1 D underneath the WE) and the high Nuclear/Cytoplasmic ratio more evident. I suggest to replace this image with a more appropriate  image of the blastematic tissue.

-line 154: I believe that the Origin of blastema cells is now ALMOST fully KNOWN, as indicated by the following sentence (lines 156-171) of this same MS (histological, experimental, molecular, genetics etc. that confirmed it is heterogenous, plastic-transdifferentiating, and lineage-specific…). The IMPORTANT-FUNDAMENTAL papers by Kragl et (Cells keep a memory..Nature 2009) and Tanaka E (The molecular and cellular choreography…..Cell 2016) and Currie P. et al. (Live imaging of axolotl……….Dev Cell 2016)…..are NOT even reported, why ?

-line 178: THE important MS by Godwin et al., 2013 Macrophages are required… PNAS) about the essential role of MACROPHAGES on newt regeneration is not considered here, why ?

-line 931:….In a study by [240]. This approach…:PLEASE CLARIFY THIS SENTENCES

-line 1014:………Since certain time…..THIS UNCLEAR SENTENCE AND THE INSERTION OF “SPACE EXPERIMENTS” IN THIS PARAGRAPH ARE UNCLEARLY INTRODUCED IN THE CONTEXT OF Environmental factors affecting regeneration. Maybe a specific SUBHEADING CLARIFYING THE INCLUSION OF THESE INTERESTING DATA WITHIN THE MS WOULD HELP THE READER TO UNDERSTAND THE RELATHIONSHIPS WITH THE OTHER CONTENT OF THIS PART OF THE MS. VARIATIONS in the orientation of the axial REGENERRATING STURCTURES -VERT COLUM AND SC-HAVE BEEN ALSO DETECTED FOR NORMALLY REGENERATING CASES.

-THE conclusion section should be CONDENSED, avoiding well known infos, also reported in the introduction etc. PLEASE emphasize the environmental effects that are a constant NEGLECTED topic in most of previous REVIEWS AND MSS DEALING WITH AMPHIBIAN REGENERATION.

Figures: ONLY 4 figures are presented but the long text of the MS would benefit of more illustrations aqnd summarizing Tables. IN GENERAL should also  be improved the labeling and the captions.

-REFERENCES: I could not check in detail the long list of refs.

-the correct citation 90 is Alibardi L. Ultrastructural observations…………..NOT Lorenzo A.

-96: align O’hara (OR O’Hara?)

-132: align Londono et

-136: align

-172: align

-177. Align

-224: align correctly

IN CONCLUSION the present MS has a GOOD potential BUT requires a shorter resume and presentation, emphasizing the new/original ideas on this topic, mentioned in the Abstract.

Comments on the Quality of English Language

n/a

Author Response

Dear Reviewer,

Reviewer 2 Report

Comments and Suggestions for Authors

The manuscript provides a well-structured and organized review of the current understanding of tail and spinal cord regeneration in urodelean amphibians (salamanders and newts). It covers various aspects, including the general course of the regeneration process, wound healing, blastema formation, spinal cord regeneration, source cells involved, molecular players (signalling pathways, transcription factors, epigenetic factors), and the influence of external factors on the regeneration process, which is an understudied area. The review highlights the unique regenerative abilities of these animals, contrasting them with mammals. It discusses the potential of this model system to provide insights into regenerative biology.

My suggestions would be:

1. The review focuses primarily on salamanders and newts. An expansion of a short review and simple comparison of other regeneration models such as zebrafish and lizards would make this study more comprehensive to this field.

2.  The content of molecular players is a little too complicated and could be simplified with more diagrams, illustrations or even tables, for better understanding to the readers.

Author Response

Dear Reviewer,

Round 2

Reviewer 2 Report

Comments and Suggestions for Authors

I am satisfied with this version of manuscript. No further suggestion.